# Global Analysis of Natural Products Biosynthetic Diversity Encoded in Fungal Genomes

**DOI:** 10.3390/jof10090653

**Published:** 2024-09-13

**Authors:** Shu Zhang, Guohui Shi, Xinran Xu, Xu Guo, Sijia Li, Zhiyuan Li, Qi Wu, Wen-Bing Yin

**Affiliations:** 1State Key Laboratory of Mycology, Institute of Microbiology, Chinese Academy of Sciences, Beijing 100101, China; fulisan12138@163.com (S.Z.); shigh@im.ac.cn (G.S.); xuxr@im.ac.cn (X.X.); 20011009gx@gmail.com (X.G.); scarlett-sijia@outlook.com (S.L.); 2Medical School, University of Chinese Academy of Sciences, Beijing 100049, China; 3Center for Quantitative Biology, Academy for Advanced Interdisciplinary Studies, Peking University, Beijing 100871, China; zhiyuanli@pku.edu.cn; 4Peking-Tsinghua Center for Life Sciences, Academy for Advanced Interdisciplinary Studies, Peking University, Beijing 100871, China

**Keywords:** fungi, gene cluster families, biosynthetic potential, biosynthetic diversity, natural products

## Abstract

Fungal secondary metabolites (SMs) represent an invaluable source of therapeutic drugs. Genomics-based approaches to SM discovery have revealed a vast and largely untapped biosynthetic potential within fungal genomes. Here, we used the publicly available fungal genome sequences from the NCBI public database, as well as tools such as antiSMASH, BIG-SLiCE, etc., to analyze a total of 11,598 fungal genomes, identifying 293,926 biosynthetic gene clusters (BGCs), which were subsequently categorized into 26,825 gene cluster families (GCFs). It was discovered that only a tiny fraction, less than 1%, of these GCFs could be mapped to known natural products (NPs). Some GCFs that only contain a single BGC internally are crucial for the biodiversity of fungal biosynthesis. Evident patterns emerged from our analysis, revealing popular taxa as prominent sources of both actual and potential biosynthetic diversity. Our study also suggests that the genus rank distribution of GCF is generally consistent with NP diversity. It is noteworthy that genera *Xylaria*, *Hypoxylon*, *Colletotrichum*, *Diaporthe*, *Nemania*, and *Calonectria* appear to possess a higher potential for SM synthesis. In addition, 7213 BGCs match possible known compound structures, and homologous gene clusters of well-known drugs can be located in different genera, facilitating the development of derivatives that share structural similarity to these drugs and may potentially possess similar biological activity. Our study demonstrated the various types of fungi with mining potential, assisting researchers in prioritizing their research efforts and avoiding duplicate mining of known resources to further explore fungal NP producers.

## 1. Introduction

Fungi produce large amounts of secondary metabolites for protection against environmental stressors and suppress competitors [1,2]. These natural products, with a variety of chemical structures and commendable biological activities, serve as essential sources for drug development, such as penicillin, anidulafungin, and griseofulvin [3,4,5]. The diversity of fungal species, particularly Ascomycota and Basidiomycota, and the accompanying diversification of biosynthetic genes and gene clusters point to an almost limitless potential for metabolic variation [6]. Recent advances in genome sequencing technology have also revealed a vast potential chemical space of hundreds of thousands of natural products yet to be discovered [6,7,8,9,10]. 

It is noteworthy that the genes responsible for secondary metabolite synthesis are typically arranged in tightly clustered regions on chromosomes [8], and the existence of these biosynthetic gene clusters (BGCs) is readily revealed in genome sequences. A BGC typically contains one or more core genes encoding biosynthetic enzymes responsible for the synthesis of the backbone of secondary metabolites. These core genes include polyketide synthases (PKSs), non-ribosomal peptide synthetases (NRPSs), their hybrid enzymes, NRPS-like enzymes, terpene synthases, as well as enzymes for the synthesis of other compounds such as indole and β-lactones [11]. PKSs are further divided into three types, and the most commonly found type in fungi is type I PKS (T1PKS) [12,13]. The type of biosynthetic enzymes encoded by the core genes in a BGC is referred to as the BGC type [14]. Recent studies have successfully identified 70,011 BGCs from 7541 prokaryotic genomes, including bacteria and archaea [15], 1,008,546 BGCs from 163,269 bacterial genomes [16], and 39,055 BGCs from 1038 ocean microbiomes [17,18]. These findings revealed that the number and kind of BGCs differed across microbial genomes and indicated that some biosynthetic pathways were unique to specific taxa. However, it has been shown that numerous sequence-similar BGCs, which synthesize compounds with extremely similar structures, have the same biological activity. If the number of BGCs were used as an indicator of biosynthetic diversity, the results would likely be overstated. Thus, many studies on how to cluster similar BGCs into gene cluster families (GCFs) have been performed and demonstrated the scientific validity of GCFs as a measure of biosynthetic diversity [16,19]. Moreover, the number of identified BGCs and GCFs was positively correlated with the number of analyzed genomes [16,19].

The fungal kingdom is one of the most phylogenetically diverse and extensive of the tree of life. It is divided into nine phyla, which comprise about 200 orders with an estimated 2.2–3.8 million species [20,21]. However, the application of GCFs to fungal genomes has been largely limited to datasets of <100 genomes from well-studied genera such as *Aspergillus*, *Fusarium*, and *Penicillium* [14,22,23,24]. Only a recent study performed a global analysis of BGCs from a dataset of 1037 fungal genomes, and nearly 12,000 fungal GCFs (from around 37,000 BGCs) were found to lack known metabolites [8], indicating that the biosynthetic potential of the fungal kingdom was significantly larger than previously assumed. Thanks to technological advances in genome sequencing, the number of fungal genomes has reached above 11,000 by the end of 2022. A >10-fold increase in the number of genomes will be better to unveil the exact potential of fungi in discovering novel secondary metabolites. Furthermore, the availability of genomes representing a broad sampling of the fungal kingdom will be helpful for discovering novel secondary metabolites and figuring out the microbial groups worthy of priority exploration via a systematic analysis of the taxonomic distribution of GCFs. 

In order to integrate the biosynthetic diversity resources of the fungal kingdom and discover certain patterns to guide researchers to mine natural products, we have initiated this work. Here, we performed a global approach based on 11,598 publicly available genomes to create a comprehensive overview of the biosynthetic diversity found across the entire fungal kingdom. We discovered 293,926 BGCs, which were arranged into 26,825 GCFs, allowing us to explore differences in biosynthetic diversity within the diverse taxonomic classes of the fungal kingdom. Comparisons within the fungal kingdom showed genus to be the most appropriate rank for comparing biosynthetic diversity across homogeneous groups. Simultaneously, we used a comprehensive dataset to explore similar gene clusters of drug molecules across multiple genera, identifying natural products (NPs) with potential biological activities. This study lays the groundwork for the systematic discovery of new compounds in the fungal kingdom and helps researchers prioritize the order of exploration of fungal species.

## 2. Materials and Methods

### 2.1. Genomic Database and BGC Prediction

We obtained 11,609 fungal genomes from NCBI on 1 October 2022, and the level of these genomes contains both complete genomes and draft assemblies. Of these, 10,598 genome files contained taxonomic information, at least at the phylum level, which were selected for analysis. Genome downloads were performed at NCBI FTP using the command line downloader aria2 (v1.36.0). A total of 20,004 natural products (NPs) found in fungi were from Natural Products Atlas (v2022.09) [25], and this database provided hierarchical clustering of the compound structures via Morgan fingerprinting and Dice similarity scoring. BGCs of known NPs of the fungal community were obtained from the MIBiG (v3.1) [26] database and were applied in subsequent analyses. Fungal genomes with annotated gbff format files were selected for BGC prediction using antiSMASH (version 6.1.1) [27]. Setting the input sequence taxon to fungi, the gene-finding tool was selected as glimmerhmm, and the BGCs predicted were compared with all known BGCs in the MIBiG (v3.1) database using the “--cb-known clusters” command. The appearance of a zip file of all output data in the folder is the criterion that the run has ended for its input genome file. The BGC files in each output folder were transferred to a new folder named after their genomes and used as the original files for clustering. According to the NCBI taxonomy tool, all the genome assembly numbers were matched to their kingdom, phylum, order, family, genus, species, and organism, and the taxonomy tsv files were outputted and used as input files for the BGCs clustering tool.

### 2.2. Clustering for GCFs and Determination of T-Values

To quantify the biosynthetic diversity of fungal genomes, we analyzed all BGCs with BiG-SLiCE (v1.1.1) [28]. A total of 110 compounds in the NPAtlas were mapped to a known BGC in the MIBiG (v3.1), allowing us to use them as an anchor for choosing our clustering threshold. Compound clusters in NPAtlas were utilized as NP clusters to compare with GCF quantities. The known fungal BGCs from MIBiG (v3.1) were used to cluster with different clustering thresholds T (100–1500), and different numbers of GCFs were obtained, respectively. Finally, we chose a threshold of 550, as it provided the most congruent agreements between the two groupings, with a v-score = 0.92 and ΔGCF = −2. Take the BGCs dataset as well as the taxonomy files as the input files of BiG-SLiCE, set the T value to 550 and other parameters as default parameters, and run the program to obtain the database files of the clustering information, as well as the visualization web pages. All the BGC files in MIBiG (v3.1) are matched to the corresponding GCFs using the query function.

### 2.3. Rarefaction Analysis

The extrapolation of a potential number of GCF values was achieved by conducting rarefaction analyses using the iNEXT R package [29]. A GCF presence/absence table (GCF-by-strain matrix) was constructed for each group considered. It was then used as ‘incidence-raw’ data in the iNEXT main function, where 500 points were inter- or extrapolated with an endpoint of 10,000 for the genomes, which is about five times more than the number of genomes in each genus in the full dataset. As the number of genomes increased, the number of GCFs and the predicted value of GCF at this point was defined as pGCF, which was used as the maximum biosynthetic potential of the genus.

### 2.4. Taxonomic Classification

The taxonomic information of the genomes comes from the NCBI taxonomy. We saved the common tree for all organisms we used and used iTOL to visualize the tree.

### 2.5. Search for Similar BGCs as Fungal Known Drugs2.4. Taxonomic Classification

BGCs of known drugs were obtained from MIBiG (v3.1). One BGC was randomly selected in each GCF as the representative BGC. A diamond database was created for these representative BGCs using Cblaster [30]. The known BGCs were used as queried files and inputted into Cblaster to search for sequence-similar BGCs. It needs the diamond database as the comparison database and requests the core genes as necessary. Using Cblaster to match the representative BGCs in the diamond database, all BGCs in their GCFs are taken out. These BGCs were used as input sequences for BiGSCAPE [31] with default parameter settings, and BGC re-clustering was performed to obtain a clustering network of BGCs. All clustering results were visualized using Cytoscape (v3.9.1) to build a network of BGCs with potential for fungal drug production.

## 3. Results

### 3.1. General Overview of GCF Resources in the Fungal Kingdom

A comprehensive set of fungi genome data (up to 1 October 2022) from the National Center for Biotechnology Information (NCBI) was collected. This dataset spans nine fungal phylum classifications, including 9427 genomes from Ascomycota, 1711 genomes from Basidiomycota, 320 genomes from Mucoromycota, and 140 genomes from other phyla. To predict BGCs, these genomic gbff files were fed into antiSMASH (v6.1.1) [27]. A total of 293,926 BGCs were identified, 268,604 of which originated in Ascomycota, 21,132 were from Basidiomycota, and 4190 were from other phyla (Figure 1A). The lengths of these BGCs span a range from 0.8 to 242 kb (Appendix A).

BiG-SLiCE is a suitable tool for clustering BGCs into GCFs for large data volumes. By inputting the sequence of BGCs along with the species classification information to BiG-SLiCE and a threshold T for clustering, these BGCs can be clustered into multiple GCFs based on the distance to the clustering model. We clustered BGCs with a certain clustering threshold T into GCFs and used the number of GCFs as an evaluation criterion for biosynthetic diversity. The threshold T determines the similarity of the BGCs within the GCFs. By utilizing the fungal BGCs from MIBiG (v3.1) [26] and the NPs contained in the Natural Products Atlas database (NPAtlas, v2022.09) [25], we performed clustering of the BGCs using different T values (ranging from 0.1 to 1.1). Finally, we obtained the GCFs with each T value (Appendix A). Concerning a recent relevant study, the absolute value of the subtraction of the number of GCFs from the number of NP clusters (ΔGCF) and the v-score was chosen as evaluation criteria [16] (Appendix A). The smaller the value of ΔGCF, the closer the number of GCFs is to the number of NP clusters. The v-score is distributed between 0 and 1, and the closer it is to 1, the more consistent the clustering labels are with the basic truth labels. It is generally considered that v-score > 0.9 means that the clustering quality is good. Ultimately, we chose 0.55 as T when ΔGCF = 2 and v-score = 0.92. Additional work demonstrates that the choice of T value has no significant effect on the intercomparison of high and low fungal biosynthetic diversity (Appendix A). The 293,926 BGCs in the fungal kingdom were clustered into 26,825 GCFs based on T = 0.55 (Figure 1A). A total of 440 BGCs from the fungal kingdom were experimentally characterized in MIBiG, and we mapped them to our GCFs. Only 165 GCFs contained known BGCs with corresponding compound structures in NPAtlas. It suggests that about 0.6% of the biosynthetic resources have been fully exploited in the fungal kingdom (Figure 1B). The output of BiG-SLiCE with a T value of 0.55 is organized as shown in Appendix A.

Analysis of the 26,825 GCFs determined that each GCF contained a distinct number of internal BGCs. A total of 1.49% of GCFs were found to contain 55.37% BGCs. These large numbers of BGCs with similar gene sequences represent a small fraction of the overall fungal biosynthetic diversity, which is widely distributed across 1003 genera. While 49.94% of GCFs contained only one BGC, this accounted for only 4.56% of the total number of BGCs (Figure 1C). This suggests that these unique BGCs that form separate GCFs may contribute to the overall biosynthetic diversity of the fungal kingdom. Within GCFs with only one BGC, 49.5% are T1PKS, and 17.5% are T1PKS&NRPS (Appendix A). It illustrates the rich diversity of polyketides in the fungal kingdom.

### 3.2. Distribution of Biosynthetic Diversity in the Fungal Kingdom

The distribution of GCFs at the phylum rank in the fungal kingdom was statistically determined, and it was found that the distribution of GCFs in the fungal kingdom was concentrated in the phyla Ascomycota and Basidiomycota. A total of 25,104 GCFs were identified in Ascomycota, representing an average of 26.8 GCFs per genome. Basidiomycota had a total of 1950 GCFs, representing an average of 9.1 GCFs per genome. Notably, these striking figures show that 98.8% of the fungal kingdom’s GCFs are shared by these two phyla (Figure 2A–C). In Ascomycota, a total of 16,655 NPs within 5335 NP clusters were recorded in the NPAtlas [25], and only 226 (0.9% of total GCFs in Ascomycota) GCFs owned the BGCs matched in MIBiG [26]. In Basidiomycota, 3647 NPs within 1453 NP clusters were recorded in NPAtlas, and only 61 (3.1% of total GCFs in Basidiomycota) GCFs owned known BGCs (Figure 2A). Currently, there are no natural products from the four clades Blastocladiomycota, Cryptomycota, Microsporidia, and Olpidiomycota in the NPAtlas database. However, the data analysis indicates that there are six, one, three, and two GCFs in these four phyla, respectively. These less-studied phyla still hold hidden biosynthetic resources that have yet to be discovered. 

The distribution ranges of GCFs across the fungal kingdom are different, some being specific to a single species while others are widespread across phyla. In this study, the extent of fungal GCF distribution was classified into seven hierarchical ranks from lowest to highest, namely, species-specific, genus-specific, family-specific, order-specific, class-specific, phylum-specific, and multi-phyla. For each genome examined, the total number of GCFs at each rank was calculated, providing an overall distribution pattern of GCFs within the fungal kingdom (Figure 2D). Throughout the fungal kingdom, the decrease in taxonomic levels (from phylum to species) is accompanied by a decrease in the number of GCFs specific at each rank per genome. The violin plot, which shows the distribution of GCFs at various ranks, indicates that there are two notable increases. These occur from species-specific to genus-specific levels and from class-specific to phylum-specific levels. It suggests that there are many shared GCFs among different species in the same genus and among different classes in the same phylum. Only 3.3 GCFs were added per genome from genus-specific level to class-specific level, suggesting that few GCFs were distributed across families or order in each genome. Thus, the GCFs in each fungal genome average 8.6 genus-specific, 7.4 multi-generic but phylum-specific, and 7.4 multi-phyla. 

The diversity of these GCFs will directly affect the diversity of fungal natural product structures. The function of enzymes encoded by the core genes in biosynthetic gene clusters (BGCs) determines the structural type of the compounds. T1PKS-type BGCs are responsible for synthesizing polyketide compounds (PKs), while NRPS-type BGCs are involved in the synthesis of non-ribosomal peptide compounds (NRPS). Sometimes, one BGC may contain both PKS and NRPS or NRPS-like enzymes, or domains of two types of enzymes may be encoded in a single gene. Among the GCFs aggregated in a single species, T1PKS was the most abundant, accounting for 49.1%, followed by T1PKS&NRPS-like, accounting for 17.9%, and NRPS-like, accounting for 12.0% (Figure 2E). This means that the structure of polyketides varies greatly between species, although some species are taxonomically closely related. Among the GCFs with the most dispersed distribution ranges, NRPS-like was the most numerous, accounting for 37.9%, followed by terpenes at 19.5% and T1PKS at 15.6% (Figure 2E). The wide distribution of these GCFs may suggest that they are more evolutionarily conserved or that their products are associated with normal physiological activities in fungi.

### 3.3. Genus-Level Analysis of Biosynthetic Diversity and Potential in the Fungal Kingdom

Based on the research methodology of recent relevant studies, we conducted a variance analysis that included each taxonomic level, from phylum to species, to ensure taxonomic level biosynthetic diversity [16]. For each rank, the variance value was computed based on the number of GCF values of immediately lower rank. The distribution of these variance values for each rank is visualized in Appendix A. There is a noticeable drop in the range of variance values for each rank, while diversity becomes highly homogeneous at the species rank. Additional statistical analysis confirmed the significance of this observation (Appendix A). Different species within a genus are likely to display uniform biosynthetic diversity, while much dissimilarity is observed between different genera belonging to the same family. Therefore, the genus is considered the most suitable taxonomic level for comparative analysis. However, this analysis relies heavily on the wide variation in the number of sequenced strains across genera. The number of GCFs per genus is recorded in Appendix A.

To overcome this bias, rarefaction analyses were performed for each genus (Figure 3A), as in previous studies [16,33]. Expanding the genome number to 10,000 (more than five times), we obtained potential GCFs (pGCFs) in each genus using the R package iNEXT [29]. As shown in Figure 3, there are 10 genera with more than 1000 pGCFs. There was no significant correlation between pGCF and genome size per genus (Appendix A). These genera in descending order are *Aspergillus*, *Fusarium*, *Xylaria*, *Hypoxylon*, *Penicillium*, *Colletotrichum*, *Talaromyces*, *Diaporthe*, *Nemania*, and *Calonectria* (a–j), and they are all from Ascomycota. The number of GCFs and pGCFs in the fungal kingdom as a whole is shown in Appendix A. These 10 genera have a combined count of 11,783 GCFs, accounting for 43.9% of all GCFs in the fungal kingdom. *Aspergillus* is considered to be the genus with the highest number of GCFs, accounting for 4130 GCFs. This represents 15.4% of all GCFs in the fungal kingdom. In addition, *Aspergillus* has a total of 6924 pGCFs, further indicating that there could potentially be a total of 2794 virtual GCFs in genomes that have not yet been characterized, as the available data are not sufficient to determine their presence. Therefore, nearly 40% of biosynthesized resources are still unexplored in the unknown genomes of *Aspergillus*. Analysis of the distribution of GCFs among these 10 genera and other genera found that from a to j, the number of genus-specific GCFs were 3048, 1606, 545, 352, 882, 846, 507, 513, 175, and 389 (Figure 3B). GCFs specific to each genus in these 10 genera accounted for 30.93% of the GCFs in the fungal kingdom.

The number of known compounds in NPAtlas [25] was counted, and genera with known compounds less than 10% of the pGCF count were considered promising candidates with high biosynthetic diversity potential. A total of six genera have known compounds below 10% of the number of pGCFs (*Xylaria*, *Hypoxylon*, *Colletotrichum*, *Diaporthe*, *Nemania*, and *Calonectria*), with the red star marked (Figure 3C). Figure 3D provides a visual representation of the number and types of GCFs for these six genera. T1PKS is the most abundant of all six genera. Among the various genera, *Calonectria* stands out for its high number of NRPS GCFs, with 120. *Colletotrichum* has the highest number of T1PKS GCFs, with a total of 739. *Xylaria*, on the other hand, has the most NRPS-like GCFs, with 139 in total. *Colletotrichum* also surpasses others with the highest number of T1PKS&NRPS GCFs, with a total of 237. There is no significant difference between these six genera in the number of terpenes, T1PKS&NRPS-like, and other types of GCFs. *Colletotrichum* is extremely rich in T1PKS and T1PKS&NRPS. In fact, 55% of the 1343 GCFs in this genus are T1PKS, and 17.6% are T1PKS&NRPS.

### 3.4. Link NP Clusters to GCFs Using Known BGCs as Anchors

Identifying BGCs with known metabolite products can link GCFs to NP clusters because BGCs in the same GCF may produce products with similar structures [8]. On the one hand, the mining process can be simplified by eliminating a significant number of repetitive and uninformative BGCs. On the other hand, for natural products or drugs that exhibit potent biological activities, BGCs within related GCFs can be mined to discover their corresponding derivatives, which may have similar structures and even enhanced activities. The NPAtlas database contains a total of 20,304 fungal natural products. Of these, only 265 BGC have already been matched with products. Using the anchor approach with known BGCs, the GCF is capable of large-scale annotation of unexplored BGCs, based on similarity to reference BGCs, identifying clusters that are likely to generate known metabolites or derivatives of knowns. Within our dataset, 94 GCFs contained known BGCs from NPAtlas, ~0.35% of the 26,825 total GCFs reported here. These families collectively include 7213 BGCs whose approximate metabolite products can now be inferred. 

To illustrate the utility of this segment of the dataset, we selected a random GCF, GCF 9748, to examine its internal BGCs. In the NP cluster, the product of the known BGC in GCF 9748 instructs us to connect the GCF to the NP cluster. GCF 9748 primarily contains T1PKS BGCs, associated with NP cluster 46, featuring 622 natural products derived from fungi. The internal members of NP cluster 46 are mainly polyketides containing hydroxybenzoic acid or its derivatives in their structure. GCF 9748 hosts 2397 BGCs, which are widely distributed across 332 genera, including *Alternaria*, *Aspergillus*, *Bipolaris*, *Colletotrichum*, *Fusarium*, and others. This pattern of biosynthetic variation within a GCF and the knowledge of their taxonomic distribution will be valuable to guide genome mining in the identification of new analogs of compounds. As shown in Figure 4A, different colors in the genes of BGCs represent biosynthesis-related domains. The compounds with the same base color as BGCs are the corresponding structures inferred from the relationship between GCF and NP cluster. The different genetic composition of these BGCs leads to the different structures of the compounds.

A large number of drug molecules have been identified in the fungal kingdom, and their biosynthetic genes and biosynthetic pathways have been resolved. By studying their sequence-similar BGCs, it is likely to obtain derivatives of these bioactive compounds. 

Penicillins and cephalosporins are two well-known classes of antimicrobial drugs containing the β-lactam ring. With the development of genomics, similarities in the BGCs of these two classes of compounds have been identified. Homologs of *pcbAB* and *pcbC*, key genes for penicillin biosynthesis, are present in the BGC of cephalosporins, but homologs of the key gene *penDE* are missing [4,34]. Therefore, mining analogs of drug compounds using sequence-similar BGCs is a possible strategy. 

We selected BGCs of several representative fungal-derived drugs, including penicillin, lovastatin, cyclosporine, mycophenolic acid, and equisetin, and mined their similar BGCs. A total of 874 BGCs similar to these known drugs were identified, and 417 (47.7%) BGCs were distributed in the 10 genera with the highest analyzed biosynthetic diversity potential (Appendix A). Seventy-seven similar penicillin BGCs are distributed in 15 genera, including 32 in *Aspergillus*, 13 in *Penicillium*, 18 in *Trichophyton*, and 14 in other genera. Only 11 similar cyclosporin BGCs were found in these BGCs, distributed in seven genera. Specific information on similar BGCs for these fungal drugs is in Appendix A.

## 4. Discussion

Fungi exhibit great potential to synthesize novel bioactive natural products, but the detailed fungal biosynthetic chemical space has not been comprehensively evaluated. Here, we conducted a global analysis of the diversity and novelty of the BGCs from 11,598 fungal genomes. Evident patterns emerged from our analysis, revealing popular taxa as prominent sources of both actual and potential biosynthetic diversity and multiple yet uncommon taxa as promising producers.

A total of 293,926 BGCs were identified, which was about eight times as many as the number reported in the previous study [8]. Among them, the unknown BGCs accounted for 99.8% and were organized into 26,825 mostly new (99.1%) GCFs, underscoring that fungi have the potential to biosynthesize compounds greatly exceeding known fungal chemical space. About 99% of novel GCFs were derived from phyla Ascomycota and Basidiomycota. This is in accordance with the known biosynthetic potential of the filamentous fungi [8,35,36,37]. At the same time, we identified taxa that were less well represented in sequence databases as being potentially useful sources of secondary metabolites. For example, several GCFs were identified from the strains of Blastocladiomycota, Cryptomycota, Microsporidia, and Olpidiomycota that have no recorded natural products in the NPAtlas database. Their potential biosynthetic capacity might be underestimated since very few genomes were contained in those groups [38]. As reported in some smaller-scale projects [8,14,22], most GCFs were species- or genus-specific, which means that expanding the dataset with species-rank sequencing of taxonomic groups is more helpful in inferring the biosynthetic potential of fungal groups.

Ascomycota is the largest among all the phyla that make up the kingdom of fungi, and it comprises about 17 classes, 115 orders, 485 families, 6540 genera, and more than 80,000 species [39]. Most genome sequencing efforts focused on this phylum, and the published genomes account for 81% of total fungal genomes. Moreover, they cover nearly all classes of Ascomycota, and four classes, Dothideomycetes, Eurotiomycetes, Saccharomycetes, and Sordariomycetes, possess over one thousand genomes. Previous studies focused on complete genome sequences revealed that the Ascomycota fungi had far greater potential to produce structurally complex, specialized metabolites than suggested by traditional bioactivity-based screening approaches [5,40,41]. In the present study, genome mining identified above 260,000 novel BGCs, and the identified GCFs account for a fraction of pGCFs. The most diverse groups of metabolites were predicted to be produced by strains of well-known and well-studied SM producers, such as *Aspergillus*, *Fusarium*, *Xylaria*, *Hypoxylon*, and *Penicillium*. Among the top 10 producers, the genera *Xylaria*, *Hypoxylon*, *Colletotrichum*, *Diaporthe*, *Nemania*, and *Calonectria* were regarded as the genera with high biosynthetic diversity potential. *Xylaria*, *Hypoxylon*, and *Nemania* belong to the order Xylariales, which represent one of the most prolific lineages of secondary metabolite producers [42]. However, the published studies generally focused on a species from these genera, and a systematic analysis of the biosynthesis potential in these genera was scarce. The published genomes were 37 from *Xylaria*, 27 from *Hypoxylon*, and 8 from *Nemania*, but the value of GCFs/per genome was higher than others. The number of identified GCFs and estimated pGCFs in *Xylaria* was about two- and six-fold of isolated bioactive compounds [43], implying that the genus *Xylaria* possessed great untapped potential in medicinal and agrochemical applications. Furthermore, we identified taxa that were less well represented in sequence databases as being potentially useful sources of secondary metabolites, such as *Monosporascus*, *Annulohypoxylon,* and *Stachybotrys*. One genome of these genera contained more than 20 GCFs, and the majority of GCFs were species-specific. Thus, as more diversified strains become sequenced, more SM diversity will be obtained in this group.

The repeated discovery of known natural products is a major challenge in the field of natural product chemistry [44]. Having the genomic capacity for the biosynthesis of secondary metabolites does not always predict the discovery of a novel chemistry [45,46]. In this study, we developed a strategy to predict the structure of possible compounds by linking GCFs to NP clusters based on known BGCs. This strategy can help avoid repetitive mining of gene clusters that generate similar compounds in various species. Once a novel natural product resulting from an unidentified gene cluster is identified, it can be quickly associated with the BGC of the entire GCF, bypassing redundant mining. On the other hand, there is a distinct possibility that these duplicate BGCs could assist in the discovery of additional active natural products. The presence of BGCs that possess the same or similar core genes in the same GCF allows for the production of structurally diverse derivatives due to differences in their gene composition, thereby expanding biosynthetic capacity and facilitating the discovery of new natural products. We applied this strategy to further investigate similar BGCs of several fungal drugs, including penicillin, lovastatin, cyclosporine, mycophenolic acid, and equisetin. As a result, we obtained 874 BGCs associated with these drugs, with more than half of these BGCs coming from the 10 genera with the highest biosynthetic potential.

## 5. Conclusions

This study demonstrates that 99.4% of the secondary metabolic potential of fungi was untapped, indicating that the potential to biosynthesize NPs greatly exceeded the known fungal chemical space. The distribution of GCFs in the fungi kingdom from phylum to species confirmed a wide variety of BGCs awaiting discovery in both under-explored and well-studied fungal taxa, particularly *Xylaria*, *Hypoxylon*, *Colletotrichum*, *Diaporthe*, *Nemania*, and *Calonectria*. In addition, we developed a strategy to reveal BCG structure–drug relationships, which will provide valuable knowledge for targeted drug discovery and guide genome mining in the identification of new bioactive natural products.

## Figures and Tables

**Figure 1 jof-10-00653-f001:**
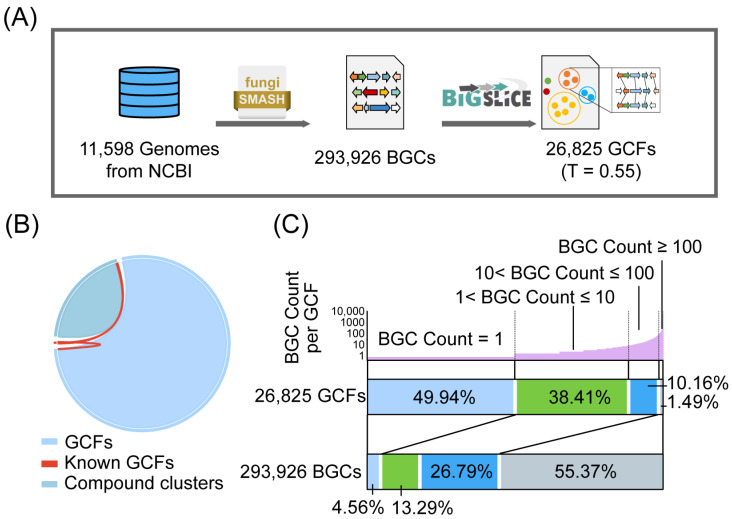
Overview of the distribution of biosynthetic diversity resources in the fungal kingdom. (**A**) Process for predicting GCFs in fungal genome files. Genome files were provided by the National Center for Biotechnology Information (NCBI), and antiSMASH (v6.1.1) [27] and BiG-SLiCE (v1.1.1) were used to predict BGCs and cluster them into GCFs, respectively. The default parameter was used in antiSMASH, and the threshold T = 0.55 was used to cluster BGCs in BiG-SLiCE. (**B**) A comparison was made between the number of GCFs obtained and known GCFs. NP clusters with a total of 6407 were reported from NPAtlas (v2022.09) [25]. The known GCFs include at least 1 BGC from MIBiG, and this BGC has a defined product in NPAtlas, 165 in total. (**C**) The number of internal BGCs per GCF and the correspondence between the number of GCFs and BGCs. The GCF is categorized into 4 types according to the number of members in the GCF: BGC Count = 1; 1 < BGC Count ≤ 10; 10 < BGC Count ≤ 100; BGC Count ≥ 10. The number of GCFs and their corresponding BGCs were analyzed using the overall data from the fungal kingdom.

**Figure 2 jof-10-00653-f002:**
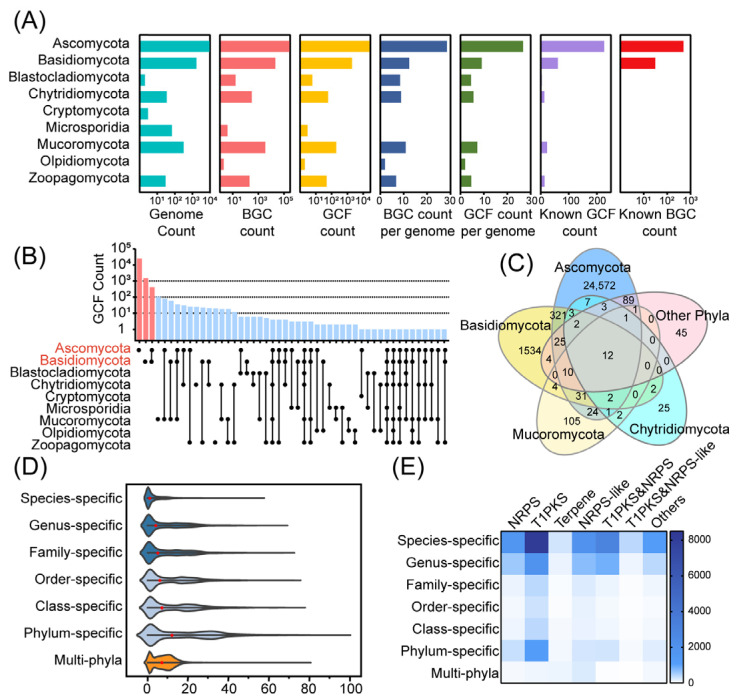
Distribution patterns of biosynthetic diversity resources in the fungal kingdom. (**A**) Statistics on the number of biosynthetic resources in the various phyla in the fungal kingdom. BiG-SLiCE GCFs were calculated with T = 0.55. (**B**) Distribution of GCF among phyla. The bar chart above shows the number of GCFs in each distribution range, and the solid black circles indicate the GCFs distributed in the phylum on the left. Pink represents the amount of GCF present only in Ascomycota and Basidiomycota. (**C**) The number of GCFs intersecting among the phyla. The top 4 phyla with the highest number of GCFs, Ascomycota, Basidiomycota, Mucoromycota, and Chytridiomycota, are listed here, categorizing Cryptomycota, Microsporidia, Olpidiomycota, Zoopagomycota as other phyla. The GCFs of these 4 phyla and other phyla are plotted in Venn diagrams, visualized by ImageGP [32]. Blue, yellow, pale yellow, pale blue and pink represent Ascomycota, Basidiomycota, Mucoromycota, Chytridiomycota and other phyla respectively. Each ellipse represents a phylum; the overlapping part of the ellipse is the number of shared GCFs, and the non-overlapping part of the ellipse is the number of GCFs unique to this phylum. (**D**) Distribution of GCF specificity at a given taxonomic level. Violin plots visualized by Origin (v 2021). The GCFs were divided into different distribution levels: species-specific; genus-specific; family-specific; order-specific; class-specific; phylum-specific; and multi-phyla. Each point in the violin means that the number of the GCFs accounts for only one sample at that taxon level in each genome, and the red point means the mean of the GCFs at this level. (**E**) The types of GCFs with different distribution ranges. Six types of GCFs common in the fungal kingdom are listed here, and several other types of GCFs, such as betalactone, indole, CDPS, and so on, which are less numerous, are combined into others. Each square indicates the number of GCFs belonging to the type below among the GCFs in the distribution range on the left, with darker colors representing larger quantities.

**Figure 3 jof-10-00653-f003:**
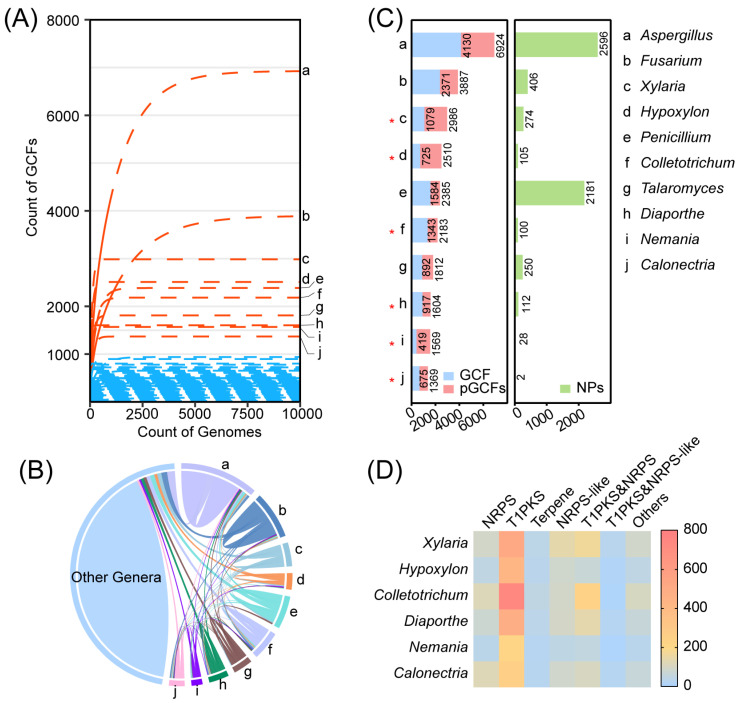
Overview of the actual and potential biosynthetic diversity of the fungal kingdom compared at the genus rank. (**A**) Dilution curves for the number of pGCFs per genus. Pink marks the genera with pGCFs over 1000, and blue marks genera with pGCFs below 1000 (BiG-SLiCE T = 0.55). Here are the top genera with the most potential GCFs: a, *Aspergillus*; b, *Fusarium*; c, *Xylaria*; d, *Hypoxylon*; e, *Penicillium*; f, *Colletotrichum*; g, *Talaromyces*; h, *Diaporthe*; i, *Nemania*; j, *Calonectria*. (**B**) Unique GCFs in the top 10 genera with the most potential GCFs. Each taxon has a distinct color, visualized with Origin 2021. (**C**) Left: potential GCFs (pGCFs) and actual number of GCFs of the top 10 most promising genera. Right: number of NPs found in the NPAtlas database [25] from each genus. Number of GCFs, pGCFs, and NPs in genera with the highest number of pGCFs (over 1000). The genera with red stars (*) mean that the number of its NPs is less than 10% of their pGCFs. (**D**) GCF types in six genera with high biosynthetic potential. The number of GCFs of a certain type goes from low to high with blue getting lighter and orange getting darker.

**Figure 4 jof-10-00653-f004:**
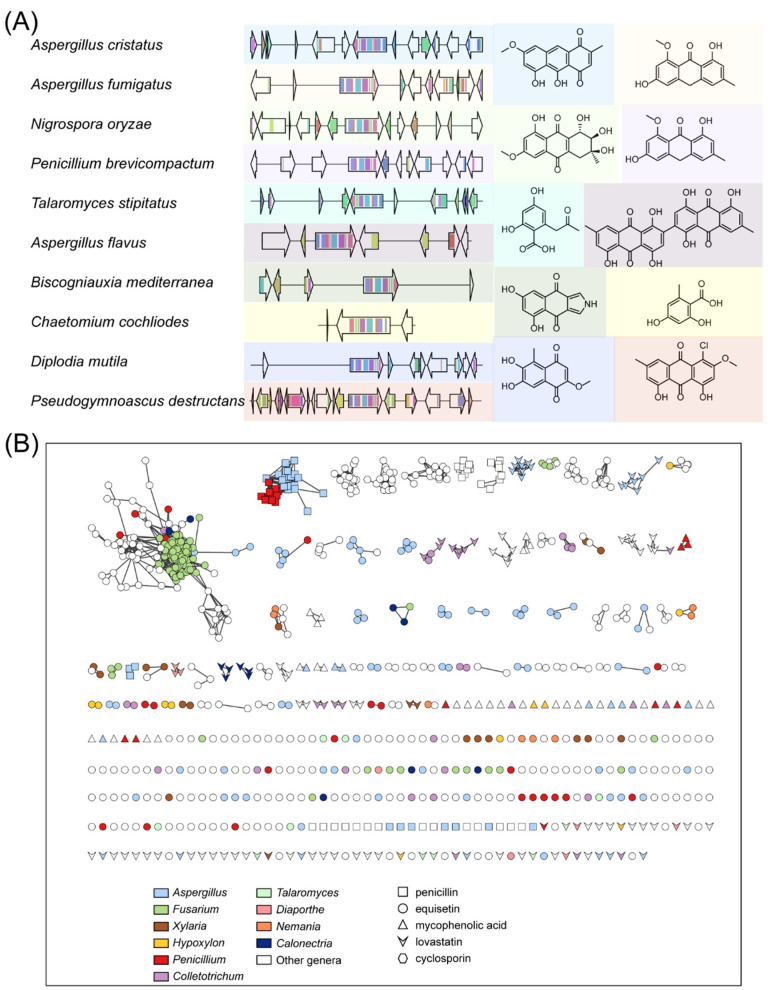
Overview of known GCFs: (**A**) Part of the BGCs in GCF 9748 with their presumed products. On the left are partial BGCs from GCF 9748, and on the right are the compound structures without known BGCs in NP cluster 46. BGCs and compound structures from the same species share a common background. In the arrows representing genes, color blocks represent the domains of biosynthesis-related proteins, and the same color represent the same domains. (**B**) Overview of BGCs similar to known drugs from fungi sources. The BGCs are similar to penicillin, equisetin, mycophenolic acid, lovastatin, and cyclosporin. Each node represents a BGC; the color represents its genus, and the shape represents its similar drugs. Cytoscape was used to visualize the BGC index.

## Data Availability

The original contributions presented in this study are included in the article/Appendix A; further inquiries can be directed to the corresponding authors.

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
