# Peer review of "Global Analysis of Natural Products Biosynthetic Diversity Encoded in Fungal Genomes"

_jof, 2024, doi:10.3390/jof10090653_

Round 1

Reviewer 1 Report

The manuscript presents the results of the analysis of all the genomic sequencing data of fungi known to date, in terms of the production of secondary metabolites. The conclusions about the species specificity of biosynthesis correlate with the known data obtained during the analysis of the Natural Products Atlas. The results of clustering can be useful to chemists and mycologists in the study of strains producing biologically active compounds. The manuscript is executed at a good analytical level and written clearly, although typos are possible in the main text.The manuscript can be recommended for publication with the wish to check the text.

The manuscript presents the results of the analysis of all the genomic sequencing data of fungi known to date, in terms of the production of secondary metabolites. The conclusions about the species specificity of biosynthesis correlate with the known data obtained during the analysis of the Natural Products Atlas. The results of clustering can be useful to chemists and mycologists in the study of strains producing biologically active compounds. The manuscript is executed at a good analytical level and written clearly, although typos are possible in the main text.The manuscript can be recommended for publication with the wish to check the text.

Author Response

The manuscript presents the results of the analysis of all the genomic sequencing data of fungi known to date, in terms of the production of secondary metabolites. The conclusions about the species specificity of biosynthesis correlate with the known data obtained during the analysis of the Natural Products Atlas. The results of clustering can be useful to chemists and mycologists in the study of strains producing biologically active compounds. The manuscript is executed at a good analytical level and written clearly, although typos are possible in the main text.The manuscript can be recommended for publication with the wish to check the text.

Response: Thank you for your careful review of our paper. We have carefully re-examined the manuscript and made corrections for spelling and formatting errors. For example, the genus names that appeared in the abstract have been modified to the italic format, and the spelling mistakes in lines 121, 176, 203, etc., have been corrected and marked in red color.

Reviewer 2 Report

The reviewer has a comment regarding the discrepancy between the manuscript title and the manuscript content. The title contains the word "compendium". As far as the reviewer knows (from available literature), a compendium is an abbreviated but comprehensive presentation of the main provisions of a field of knowledge, for example, there is a compendium of drugs, etc. It would be logical to assume (based on the manuscript title) that this provides brief information on the potential biosynthetic activity of fungi of various taxonomic groups. However, this is not the case. In the manuscript, the authors describe the process and results of the analysis of 11,598 fungal genomes and indicate that the biosynthetic potential of fungi is enormous. That is, this manuscript is not a reference material. Apparently, the authors should change the title of the manuscript. 

Lines 83-91 in the Introduction section describe the results obtained by the authors; the reviewer believes that the authors should have formulated a hypothesis for their study instead. At the moment, after reading the Introduction section, there is no understanding of the aim of the study that the authors wanted to achieve.

The Materials and Methods section is well described; various bioinformatics methods are widely used here.

The Results section is described clearly and in detail; however, the reviewer has a comment on this section. The reviewer thinks that the information content of the figures from the Supplementary Material is remarkably high, and that having these figures in the body of the manuscript would not be superfluous. Is it possible to do this?

Author Response

The reviewer has a comment regarding the discrepancy between the manuscript title and the manuscript content. The title contains the word "compendium". As far as the reviewer knows (from available literature), a compendium is an abbreviated but comprehensive presentation of the main provisions of a field of knowledge, for example, there is a compendium of drugs, etc. It would be logical to assume (based on the manuscript title) that this provides brief information on the potential biosynthetic activity of fungi of various taxonomic groups. However, this is not the case. In the manuscript, the authors describe the process and results of the analysis of 11,598 fungal genomes and indicate that the biosynthetic potential of fungi is enormous. That is, this manuscript is not a reference material. Apparently, the authors should change the title of the manuscript.

Response: Thank you very much for your valuable suggestions. The title has been changed to “Global analysis of natural products biosynthetic diversity encoded in fungal genomes”

Comment 2. Lines 83-91 in the Introduction section describe the results obtained by the authors; the reviewer believes that the authors should have formulated a hypothesis for their study instead. At the moment, after reading the Introduction section, there is no understanding of the aim of the study that the authors wanted to achieve.

Response: This study aims to explore the distribution patterns by integrating known fungal biosynthetic resources, and guide researchers to mine natural products through the patterns we discovered. This information has been added in lines 84 and 85 to make the aim of this study more clearly.

Comment 3. The Materials and Methods section is well described; various bioinformatics methods are widely used here. The Results section is described clearly and in detail; however, the reviewer has a comment on this section. The reviewer thinks that the information content of the figures from the Supplementary Material is remarkably high, and that having these figures in the body of the manuscript would not be superfluous. Is it possible to do this?

Response: Thank you for the good suggestions and positive comments about the Supplementary Materials. After careful reading, we confirm that the crucial information in the supplementary Material has been described in the Results section. It may require reorganizing the manuscript structure. Therefore, we prefer to keep the existing manuscript organization.

Reviewer 3 Report

Comments on Manuscript jof-3175235

The authors of the manuscript "Compendium of natural products biosynthetic diversity encoded in fungal genomes" employed the global approach based on 11,598 publicly available genomes to create a comprehensive overview of the biosynthetic diversity found across the entire fungal kingdom, giving the groundwork for the systematic discovery of new compounds in the fungal kingdom, thus helping the researchers to prioritize the order of exploration of fungal species.

The article is theoretical in its main features, and the authors covered the chosen field of investigation.

Author Response

The authors of the manuscript "Compendium of natural products biosynthetic diversity encoded in fungal genomes" employed the global approach based on 11,598 publicly available genomes to create a comprehensive overview of the biosynthetic diversity found across the entire fungal kingdom, giving the groundwork for the systematic discovery of new compounds in the fungal kingdom, thus helping the researchers to prioritize the order of exploration of fungal species. The article is theoretical in its main features, and the authors covered the chosen field of investigation.

Response: Thanks for your careful review of our manuscript and positive comments of this work.

Reviewer 4 Report

This manuscript describes an interesting study on the occurrence and distribution of secondary metabolite genes in fungal genomes.  The focus is on biosynthetic gene clusters as potential sources of natural products with possible medical importance.  

The analytical parts of this study are straightforward, using known programs to analyze genomes and gene distributions,  across a large number of known genome sequences.  

A number of questions remain that should be addressed.  Given the diversity of fungal genomes,  they are unevenly distributed across this study:  >9000 genomes from Ascomycetes, >1700 from Basidiomycota, 320 from Mucoromycota and 140 from other phyla.  How does this uneven distribution effect the outcome of the analysis?  Can this be addressed?

Identification of similarity in gene sequence is important in identifying function.  What percent identity or similarity was used to as a criterion for inclusion in a cluster?  Is the identification of gene cluster biased by sequences that are unusual but for which the protein product has the same function?

Certain genera have a high number of NRPS GCFs.  Is this related to the environment in which they are found?

see above

Author Response

 This manuscript describes an interesting study on the occurrence and distribution of secondary metabolite genes in fungal genomes. The focus is on biosynthetic gene clusters as potential sources of natural products with possible medical importance. The analytical parts of this study are straightforward, using known programs to analyze genomes and gene distributions, across a large number of known genome sequences. A number of questions remain that should be addressed.

Comment 1. Given the diversity of fungal genomes, they are unevenly distributed across this study: >9000 genomes from Ascomycetes, >1700 from Basidiomycota, 320 from Mucoromycota and 140 from other phyla. How does this uneven distribution effect the outcome of the analysis? Can this be addressed?

Response: Thank you for pointing this out. We also took into account the differences in sample size among phyla. For example, we additionally calculated the average number of BGCs and GCFs per genome in each phylum (Fig 2A), and the results presented a similar trend to the data for the total number statistics. To find a suitable taxonomic level for assessment of biosynthetic diversity, we conducted a variance analysis that included each taxonomic level, from phylum to species, and found different species within a genus are likely to display uniform biosynthetic diversity, while much dissimilarity is observed between different genera belonging to the same family (Fig S5 and S6). Therefore, some comparative analyses were performed at the genus level not at the phylum level. Moreover, to overcome this bias, rarefaction analyses were performed for each genus (Figure 3A), which expanding the genome number to 10,000 (more than 5 times) and predict potential GCFs (pGCFs) in each genus. The detailed information has provided in section 3.3.

Comment 2. Identification of similarity in gene sequence is important in identifying function. What percent identity or similarity was used to as a criterion for inclusion in a cluster? Is the identification of gene cluster biased by sequences that are unusual but for which the protein product has the same function?

Response: The process of clustering biosynthetic genes into gene cluster families utilized the BiG-SLICE algorithm. By inputting the sequence of BGCs along with the species classification information to BiG-SLiCE, these BGCs can be clustered into multiple GCFs based on the distance to the clustering model. The clustering threshold T determines by utilizing the fungal BGCs from MIBiG (version 3.1) and the NPs contained in the Natural Products Atlas database (NPAtlas, version 2022.09). In addition, the absolute value of the subtraction of the number of GCFs from the number of NP clusters (ΔGCF) and the v-score were also chosen as evaluation criteria (Figure S2). These processes mean the BGCs and GCF were obtained using a more complex computational approach than clustering by sequence similarity. Thus, the bias that induced by sequence similarity can be ignorance in the present study. The detailed information about the criterion for clustering has been provided in Section 3.1.

Comment 3. Certain genera have a high number of NRPS GCFs. Is this related to the environment in which they are found?

Response: There are certain genera that have a large number of NRPS in their genomes, but we did not find evidence that this was related to their environments. Because the species or strains in the same genus often origin from multiple environments, but their GCF resources do not differ significantly.

Round 2

Reviewer 3 Report

Comment on jof-3175235

The revised version "Compendium of natural products biosynthetic diversity encoded in fungal genomes" is acceptable for publishing.

Comment on jof-3175235

The revised version "Compendium of natural products biosynthetic diversity encoded in fungal genomes" is acceptable for publishing.